# Bayesian filtering unifies adaptive and non-adaptive neural network optimization methods

**Laurence Aitchison**
Department of Computer Science
University of Bristol
Bristol, UK, BS8 1UB
`laurence.aitchison@bristol.ac.uk`

## Abstract

We formulate the problem of neural network optimization as Bayesian filtering, where the observations are backpropagated gradients. While neural network optimization has previously been studied using natural gradient methods which are closely related to Bayesian inference, they were unable to recover standard optimizers such as Adam and RMSprop with a root-mean-square gradient normalizer, instead getting a mean-square normalizer. To recover the root-mean-square normalizer, we find it necessary to account for the temporal dynamics of all the other parameters as they are optimized. The resulting optimizer, AdaBayes, adaptively transitions between SGD-like and Adam-like behaviour, automatically recovers AdamW, a state of the art variant of Adam with decoupled weight decay, and has generalisation performance competitive with SGD.

## 1  Introduction and Background

The canonical non-adaptive neural network optimization method is vanilla stochastic gradient descent (SGD) with momentum which updates parameters by multiplying the exponential moving average gradient, $\langle g(t) \rangle$, by a learning rate, $\eta_{\mathrm{SGD}}$,

$$\Delta w_{\mathrm{SGD}}(t) = \eta_{\mathrm{SGD}} \frac{\langle g(t) \rangle}{\text{minibatch size}}. \tag{1}$$

Here, we divide by the minibatch size because we define $g(t)$ to be the gradient of the *summed* loss, whereas common practice is to use the gradient of the *mean* loss. Following the convention established by Adam (Kingma & Ba, 2015), $\langle g(t) \rangle$, is a debiased exponential moving average,

$$m(t) = \beta_1 m(t-1) + (1 - \beta_1) \, g(t) \qquad\qquad \langle g(t) \rangle = \frac{m(t)}{1 - \beta_1^t}. \tag{2}$$

where $g(t)$ is the raw minibatch gradient, and $\beta_1$ is usually chosen to be 0.9. These methods typically give excellent generalisation performance, and as such are used to train many state-of-the-art networks (e.g. ResNet (He et al., 2016), DenseNet (Huang et al., 2017), ResNeXt (Xie et al., 2017)).

Adaptive methods change the learning rates as a function of past gradients. These methods date back many years (e.g. vario-eta Neuneier & Zimmermann, 1998), and many variants have recently been developed, including AdaGrad (Duchi et al., 2011), RMSprop (Hinton et al., 2012) and Adam (Kingma & Ba, 2015). The canonical adaptive method, Adam, normalises the exponential moving average gradient by the root mean square of past gradients,

$$\Delta w_{\mathrm{Adam}}(t) = \eta_{\mathrm{Adam}} \frac{\langle g(t) \rangle}{\sqrt{\langle g^2(t) \rangle}}. \tag{3}$$

where,

$$v(t) = \beta_2 v(t-1) + (1-\beta_2)\, g^2(t) \qquad\qquad \langle g^2(t)\rangle = \frac{v(t)}{1-\beta_2^t}, \tag{4}$$

and where $\beta_2$ is typically chosen to be 0.999. These methods are often observed to converge faster, and hence may be used on problems which are more difficult to optimize (Graves, 2013), but can give worse generalisation performance than non-adaptive methods (Keskar & Socher, 2017; Loshchilov & Hutter, 2017; Wilson et al., 2017; Luo et al., 2019).

Obtaining a principled theory of adaptive optimization is important, as it should enable us to develop improved optimizers. Here we formulated Bayesian inference as an optimization problem (Puskorius & Feldkamp, 1991; Sha et al., 1992; Puskorius & Feldkamp, 1994, 2001; Feldkamp et al., 2003; Ollivier, 2017), and carefully considered how optimization of the other parameters influences the optimum of our parameter of interest. We were able to recover the standard root-mean-square normalizer for RMSprop and Adam, and we recovered a state-of-the-art variant of Adam with "decoupled" weight decay (Loshchilov & Hutter, 2017). We hope that by pursuing our dynamical Bayesian approach further it will be possible to develop improved adaptive optimization algorithms.

## 2 Related work

Previous work has considered the relationships between adaptive stochastic gradient descent methods and variational online Newton (VON), which is very closely related to natural gradients (Khan & Lin, 2017; Khan et al., 2017, 2018) and Bayes (Ollivier, 2017). Critically, this work found that direct application of VON/Bayes gives a sum-squared normalizer, as opposed to a root-mean-squared normalizer as in Adam and RMSProp. In particular, see Eq. 7 in Khan et al. (2018), which gives the Variational-online Newton (VON) updates, and includes a mean-squared gradient normalizer. To provide a method that matches Adam and RMSProp more closely, they go on to provide an ad-hoc modification of the VON updates, with a root-mean-square normalizer, saying "Using ... an additional modification in the VON update, we can make the VON update very similar to RMSprop. Our modification involves taking the square-root over $\mathbf{s}(t+1)$ in Eq. (7)". In contrast, our approach gives the root-mean-square normalizer directly, without any ad-hoc modifications, and automatically recovers decoupled weight decay (Loshchilov & Hutter, 2017) which is not recovered by VON (again, see Eq. 7 in Khan et al., 2018).

An alternative view on these results is given by considering equivalence of online natural gradients and Kalman filtering (Ollivier, 2017). Through this equivalence, they have the same issues as in (Khan & Lin, 2017; Khan et al., 2017, 2018): having a mean-square rather than root-mean-square form for the gradient normalizer. Further, note that they do consider a "fading memory" approach, they "multiply the log-likelihood of previous points by a forgetting factor $(1-\lambda_t)$ before each new observation. This is equivalent to an additional step $\mathbf{P}_{t-1} \to \mathbf{P}_{t-1}/(1-\lambda_t)$ in the Kalman filter, or to the addition of an artificial process noise $\mathbf{Q}_t$ proportional to $\mathbf{P}_{t-1}$", where $\mathbf{P}_{t-1}$ is their posterior covariance matrix. Critically, their "artificial process noise ... proportional to $\mathbf{P}_{t-1}$" again gives a mean-square form for the gradient normalizer (see Appendix C for details). In contrast, we give an alternative motivation for the introduction of *fixed* process noise, and show that fixed process noise recovers the root-mean-square gradient normalizer in Adam.

## 3 Methods

Here, we set up the problem of neural network optimization as Bayesian inference. Typically, when performing Bayesian inference, we would like to reason about correlations in the full posterior over all parameters jointly (Fig. 1A). However, neural networks have so many parameters that reasoning about correlations is intractable: instead, we are forced to work with factorised approximate posteriors. To understand the effects of factorised approximate posteriors, consider the $i$th parameter. The current estimate of the other parameters, $\boldsymbol{\mu}_{-i}(t)$ changes over time, $t$, as they are optimized. As there are correlations in the posterior, the optimal value for the $i$th parameter, $w_i^*(t)$, conditioned on the current setting of the other parameters, $\boldsymbol{\mu}_{-i}(t)$ also changes over time (Fig. 1B),

$$w_i^*(t) = \arg\max_{w_i} \mathcal{L}\left(w_i, \boldsymbol{\mu}_{-i}(t)\right). \tag{5}$$

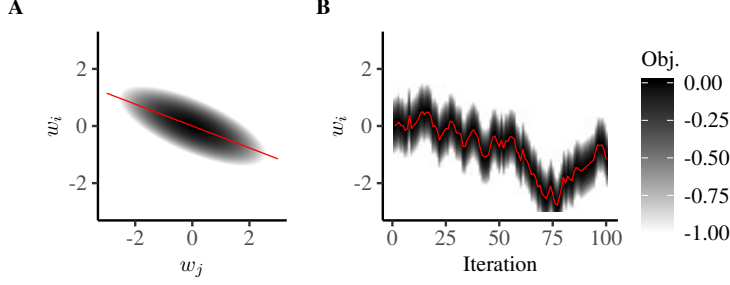

Figure 1: A schematic figure showing correlation-induced dynamics. **A** The objective function (usually equivalent to a posterior over the parameters) induces correlations between the parameter of interest, $w_i$, and other parameters (here represented by $w_j$). The red line displays the optimal value for $w_i$ as a function of $w_j$ or time. **B** The other parameters (including $w_j$) change over time as they are also being optimized, implying that the optimal value for $w_i$ changes over iterations.

where $\mathcal{L}\left(w_i, \boldsymbol{\mu}_{-i}(t)\right)$ is the objective, with the $i$th parameter set to $w_i$ and the other parameters set to $\boldsymbol{\mu}_{-i}(t)$. As such, to form optimal estimates, we need to reason about changes over time in the optimal setting for that parameter, $w_i^*(t)$. If we knew the full, correlated posterior in Fig. 1A, then we could compute the change in the $i$th parameter from the change in all the other parameters. However, in our case, the correlations are unknown, so the best we can do is to say that the new optimal value for the $i$th parameter will be close to — but slightly different from — the current optimal value, and these changes in the optimal value (Fig. 1B) constitute stochastic-dynamics that are implicitly induced by our choice of a factorised approximate posterior.

The Bernstein von-Mises theorem (Van der Vaart, 2000) indicates that in a typical setting with a large training set the log-posterior, which is taken to be the objective, is asymptotically quadratic,

$$\mathcal{L}(\mathbf{w}) = -\tfrac{1}{2}\mathbf{w}^T \mathbf{H} \mathbf{w} + c \tag{6}$$

where we take the mode to be at $\mathbf{w} = \mathbf{0}$ without loss of generality, and we verify the approximation empirically even for minibatches in Appendix A. The gradient is

$$\frac{\partial}{\partial \mathbf{w}}\mathcal{L}\left(\mathbf{w}\right) = -\mathbf{H}\mathbf{w} \tag{7}$$

Thus, the full batch gradient with respect to the $i$th weight, when all the other parameters, $\mathbf{w}_{-i}$ are set to the current estimate, $\boldsymbol{\mu}_{-i}(t)$, is

$$\frac{\partial}{\partial w_i}\mathcal{L}\left(w_i, \boldsymbol{\mu}_{-i}(t)\right) = -H_{ii}w_i - \mathbf{H}_{-i,i}^T \boldsymbol{\mu}_{-i}(t) \tag{8}$$

where $\mathbf{H}_{-i,i}^T$ is the $i$th column of the Hessian, omitting the $ii$th element. The optimal value for the $i$th parameter can be found by finding the weight for which the gradient of the objective is zero,

$$w_i^*(t) = -\frac{1}{H_{ii}}\mathbf{H}_{-i,i}^T \boldsymbol{\mu}_{-i}(t). \tag{9}$$

By substituting this value for $w_i^*(t)$ into Eq. 8, we get the gradient of the objective in terms of $w_i^*(t)$,

$$\frac{\partial}{\partial w_i}\mathcal{L}\left(w_i, \boldsymbol{\mu}_{-i}(t)\right) = H_{ii}\left(w_i^*(t) - w_i\right), \tag{10}$$

Critically, the data we actually measure is the gradient of the objective for a minibatch at the current estimate of the parameter, $w_i = \mu_i$. The minibatch gradients are Gaussian as we verify empirically in Appendix B, because the gradient for each datapoint is IID, and there are many (128) datapoints in each training batch, so the CLT applies. Following the standard approach in this line of work (Khan & Lin, 2017; Zhang et al., 2017; Khan et al., 2017, 2018), the Fisher Information (FI) can be used to identify the variance,

$$\mathrm{P}\left(g_i(t)|w_i^*(t)\right) = \mathcal{N}\left(H_{ii}\left(w_i^*(t) - \mu_i(t)\right), H_{ii}\right). \tag{11}$$

Improving the commonly used FI approximation is an important avenue for future work, but not our focus here. Nonetheless, we verify its effectiveness empirically in Appendix A.

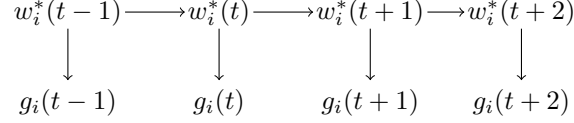

$$w_i^*(t-1) \longrightarrow w_i^*(t) \longrightarrow w_i^*(t+1) \longrightarrow w_i^*(t+2)$$
$$\downarrow \qquad\qquad \downarrow \qquad\qquad \downarrow \qquad\qquad \downarrow$$
$$g_i(t-1) \qquad g_i(t) \qquad g_i(t+1) \qquad g_i(t+2)$$

Figure 2: Graphical model under which we perform inference.

Now that Eq. (11) gives us a likelihood, we need a prior over $w_i^*(t)$. Eq. (9) shows us that the dynamics of $w_i^*(t)$ are governed by the dynamics of our estimates, $\boldsymbol{\mu}_{-i}(t)$, as they are optimized, and this optimization is a complex stochastic process. Nonetheless, we know that the parameter updates are governed by gradients, and that gradients are Gaussian (Eq. 11). Thus, it seems reasonable to take the dynamics of $\boldsymbol{\mu}_{-i}(t)$ to be Gaussian,

$$\mathrm{P}\left(\boldsymbol{\mu}_{-i}(t+1)|\boldsymbol{\mu}_{-i}(t)\right) = \mathcal{N}\left(\left(1 - \tfrac{\eta^2}{2\sigma^2}\right)\boldsymbol{\mu}_{-i}(t), \mathbf{Q}_{-i}\right). \tag{12}$$

Thus, the dynamics for the $i$th optimal weight become,

$$\mathrm{P}\left(w_i^*(t+1)|w_i^*(t)\right) = \mathcal{N}\left(\left(1 - \tfrac{\eta^2}{2\sigma^2}\right)w_i^*(t), \eta^2\right) \tag{13}$$

where,

$$\eta^2 = \mathbf{H}_{-i,i}^T \mathbf{Q}_{-i} \mathbf{H}_{-i,i}. \tag{14}$$

and where $\sigma^2$ will become the prior variance of the parameters. Combined, Eq. (11) and Eq. (13) define a stochastic linear dynamical system for each parameter separately, where the optimal weight, $w_i^*(t)$, is the latent variable, and the gradients, $g_i(t)$ are the observations (Fig. 2). As such, in the remainder of this section, we will drop parameter indices, so $w^*(t) = w_i^*(t)$ and $g(t) = g_i(t)$, to simplify notation.

As the dynamics and likelihood are Gaussian, the Kalman filter priors and posteriors are,

$$\mathrm{P}\left(w^*(t)|g(t-1), \ldots, g(1)\right) = \mathcal{N}\left(\mu_{\mathrm{prior}}(t), \sigma_{\mathrm{prior}}^2(t)\right), \tag{15a}$$

$$\mathrm{P}\left(w^*(t)|g(t), \ldots, g(1)\right) = \mathcal{N}\left(\mu_{\mathrm{post}}(t), \sigma_{\mathrm{post}}^2(t)\right), \tag{15b}$$

where we evaluate the gradient, $g(t) = g_i(t)$ at $\mu_i(t) = \mu_{\mathrm{prior}}(t)$, and where the updates for $\mu_{\mathrm{prior}}(t)$ and $\sigma_{\mathrm{prior}}^2(t)$ can be computed from Eq. (13),

$$\mu_{\mathrm{prior}}(t) = \left(1 - \tfrac{\eta^2}{2\sigma^2}\right)\mu_{\mathrm{post}}(t-1), \tag{16a}$$

$$\sigma_{\mathrm{prior}}^2(t) = \left(1 - \tfrac{\eta^2}{2\sigma^2}\right)^2 \sigma_{\mathrm{post}}^2(t-1) + \eta^2. \tag{16b}$$

And the updates for $\mu_{\mathrm{post}}(t)$ and $\sigma_{\mathrm{post}}^2(t)$ come from applying Bayes theorem (Appendix D), with the likelihood given by Eq. (11). Again, following the standard approach in this line of work (Khan & Lin, 2017; Zhang et al., 2017; Khan et al., 2017, 2018), we approximate $H_{ii}$ using the squared gradient (again, while improving this approximation is an important avenue for future work, we empirically verify its effectiveness in Appendix A)

$$\sigma_{\mathrm{post}}^2(t) = \frac{1}{\frac{1}{\sigma_{\mathrm{prior}}^2(t)} + H_{ii}} \approx \frac{1}{\frac{1}{\sigma_{\mathrm{prior}}^2(t)} + g^2(t)}, \tag{17a}$$

$$\mu_{\mathrm{post}}(t) = \mu_{\mathrm{prior}}(t) + \sigma_{\mathrm{post}}^2(t)g(t). \tag{17b}$$

The full updates are now specified by iteratively applying Eq. (16) and Eq. (17).

Next, we make two minor modifications to the updates for the mean, to match current best practice for optimizing neural networks. First, we allow more flexibility in weight decay, by replacing the $\eta^2/(2\sigma^2)$ term in Eq. (16a) with a new parameter, $\lambda$. Second, we incorporate momentum, by using an exponential moving average gradient, $\langle g(t) \rangle$, instead of the raw minibatch gradient in Eq. (17b). In combination, the updates for the mean become,

$$\mu_{\mathrm{prior}}(t) = (1 - \lambda)\,\mu_{\mathrm{post}}(t-1), \tag{18a}$$

$$\mu_{\mathrm{post}}(t) = \mu_{\mathrm{prior}}(t) + \sigma_{\mathrm{post}}^2(t)\langle g(t) \rangle. \tag{18b}$$

Our complete Bayesian updates are now given by using Eq. (18) to update $\mu_{\mathrm{prior}}$ and $\mu_{\mathrm{post}}$, and using Eq. (16b) and Eq. (17a) to update $\sigma_{\mathrm{prior}}^2$ and $\sigma_{\mathrm{post}}^2$ (see Algo. 1).

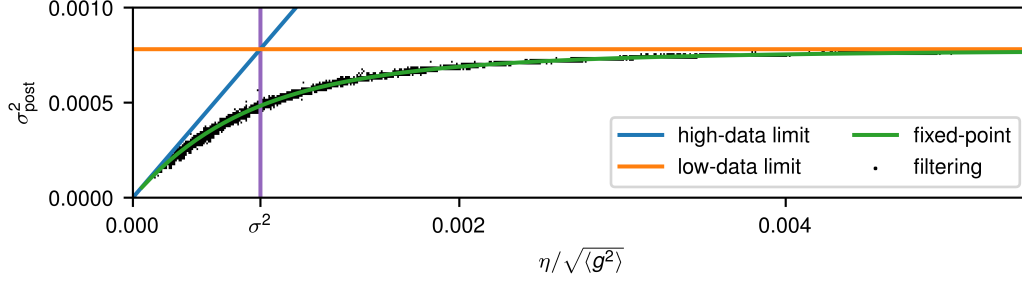

Figure 3: The learning rate for AdaBayes (points) compared against the predicted fixed-point value (green line), $\sigma_{\text{post}}^2$. The plot displays the low-data limit (orange line), which is valid when the value on the x-axis, $\eta/\sqrt{\langle g^2 \rangle}$, is much greater than $\sigma^2$ (purple line), and the high-data limit (blue line), which is valid when the value on the x-axis is much smaller than $\sigma^2$ (purple line).

### 3.1 Philosophical note

One might worry that inferring a distribution over the optimal weight, $w_i^*(t)$, does not make sense, because we could in principle find it directly by fixing all the other parameters and computing the full loss for all values of $w_i$. However, in practice this is too expensive, especially if it must be repeated for all parameters. Instead, it is necessary to summarise our tractably-computable beliefs about $w_i^*(t)$ as a probability distribution. This approach mirrors "a sermon on reality vs. models" in Jaynes (2003). He considers specifying the exact physical state of an urn of balls, shaking it, and drawing a ball from the urn. In principle, we can simulate the deterministic shaking process and thus compute the ball that would be drawn out of the urn. In practice, this computation is far beyond the capabilities of our current computers. But all is not lost, as we can still use arguments based on randomisation to make standard probabilistic judgements concerning the probability of drawing a ball of a given color.

### 3.2 AdaBayes recovers SGD and Adam

To understand how AdaBayes relates to previous algorithms (SGD and Adam), we plotted the AdaBayes learning rate, $\sigma_{\text{post}}^2$ against the Adam learning rate, $\eta/\sqrt{\langle g^2 \rangle}$ (Fig. 3, points) for the ResNet-34 considered later. We found that for high values of $\langle g^2 \rangle$, corresponding to large values of the Fisher Information, the AdaBayes learning rate closely matched the Adam learning rate (Fig. 3, blue line). In contrast, as the value of $\langle g^2 \rangle$ decreased, corresponding to smaller values of the Fisher Information, we found that the AdaBayes learning rate became constant, mirroring standard SGD (Fig. 3, orange line). Thus, Fig. 3 empirically establishes that AdaBayes converges to SGD in the low data (Fisher-Information) limit, and Adam in the high data limit. Furthermore, the tight vertical spread of points in Fig. 3 indicates that, in practice, the AdaBayes value of $\sigma_{\text{post}}^2$ is largely determined by the Fisher-Information, $\langle g^2 \rangle$, thus raising the question of whether we can obtain better understanding of the relationship between $\langle g^2 \rangle$ and $\sigma_{\text{post}}^2$. Indeed, such an understanding is possible, if we consider the fixed point of the $\sigma_{\text{post}}^2$ updates (Eq. 16b and 17a). To obtain the fixed-point, we substitute the update for $\sigma_{\text{post}}^2$ (Eq. 17a) into the update for $\sigma_{\text{prior}}^2$ (Eq. 16b), and neglect small terms (see Appendix E), which tells us that the fixed-point $\sigma_{\text{post}}^2$ is given by the solution of a quadratic equation,

$$0 \approx \sigma^2 \left( \frac{1}{\sigma_{\text{post}}^2} \right)^2 - \frac{1}{\sigma_{\text{post}}^2} - \frac{\langle g^2 \rangle \sigma^2}{\eta^2}. \tag{19}$$

Solving for $1/\sigma_{\text{post}}^2$, we obtain,

$$\frac{1}{\sigma_{\text{post}}^2} \approx \frac{1}{2\sigma^2} \left( 1 + \sqrt{1 + 4 \left( \frac{\sigma^2}{\eta/\sqrt{\langle g^2 \rangle}} \right)^2} \right). \tag{20}$$

We confirmed the fixed-point indeed matches the empirically measured AdaBayes learning rates by plotting the fixed-point predictions in Fig. 3 (green line). Importantly, the fixed-point expression

**Algorithm 1** AdaBayes

$\eta \quad \leftarrow \eta_{\text{Adam}}$
$\sigma^2 \quad \leftarrow \eta_{\text{SGD}}/\text{minibatch size}$
$\sigma^2_{\text{prior}} \leftarrow \sigma^2$
**while** not converged **do**
$\quad g \quad \leftarrow \nabla \mathcal{L}_t(\mu)$
$\quad m \quad \leftarrow \beta_1 m + (1 - \beta_1) g$

$\quad \langle g \rangle \leftarrow m/(1 - \beta_1^t)$

$\quad \sigma^2_{\text{post}} \leftarrow \frac{1}{\sigma^{-2}_{\text{prior}} + g^2}$

$\quad \sigma^2_{\text{prior}} \leftarrow \left(1 - \frac{\eta^2}{2\sigma^2}\right)^2 \sigma^2_{\text{post}} + \eta^2$
$\quad \mu \quad \leftarrow (1 - \lambda)\mu + \sigma^2_{\text{post}} \langle g \rangle$
**end while**

---

**Algorithm 2** AdaBayes-FP

1: $\eta \quad \leftarrow \eta_{\text{Adam}}$
2: $\sigma^2 \quad \leftarrow \eta_{\text{SGD}}/\text{minibatch size}$

4: **while** not converged **do**
5: $\quad g \quad \leftarrow \nabla \mathcal{L}_t(\mu)$
6: $\quad m \quad \leftarrow \beta_1 m + (1 - \beta_1) g$
7: $\quad v \quad \leftarrow \beta_2 v + (1 - \beta_2) g^2$
8: $\quad \langle g \rangle \leftarrow m/(1 - \beta_1^t)$
9: $\quad \langle g^2 \rangle \leftarrow v /(1 - \beta_2^t)$
10:

11: $\quad \sigma^2_{\text{post}} \leftarrow \left(\frac{1}{2\sigma^2} + \sqrt{\frac{1}{4\sigma^4} + \frac{\langle g^2 \rangle}{\eta^2}}\right)^{-1}$

12: $\quad \mu \quad \leftarrow (1 - \lambda)\mu + \sigma^2_{\text{post}} \langle g \rangle$
13: **end while**

---

merely helps understand a result that we established empirically. Finally, this close match means that we can define another set of updates, AdaBayes-FP, where we set $\sigma^2_{\text{post}}$ directly to the fixed-point value, using Eq. (20), rather than using the full AdaBayes updates given by Eq. (16b) and Eq. (17a).

### 3.2.1 Recovering SGD in the low-data limit

In the low-data regime where $\eta/\sqrt{\langle g^2 \rangle} \gg \sigma^2$, the empirically measured AdaBayes learning rate, $\sigma^2_{\text{post}}$, becomes constant (Fig. 3; orange line), so the AdaBayes updates (Eq. 18b) become approximately equivalent to vanilla SGD (Eq. 1). To understand this convergence, we can leverage the fixed-point expression in Eq. (20) which accurately models empirically measured learning rates,

$$\lim_{\langle g^2 \rangle \to 0} \sigma^2_{\text{post}} \approx \sigma^2, \tag{21}$$

We can leverage this equivalence to set $\sigma^2$ using standard values of the SGD learning rate,

$$\sigma^2 = \frac{\eta_{\text{SGD}}}{\text{minibatch size}}. \tag{22}$$

Setting $\sigma^2$ in this way would suggest $\sigma^2 \sim 0.001$[1], as $\eta_{\text{SGD}} \sim 0.1$, and the minibatch size $\sim 100$. It is important to sanity check that this value of $\sigma^2$ corresponds to Bayesian filtering in a sensible generative model. In particular, note that $\sigma^2$ is the variance of the prior over $w_i$, and as such $\sigma^2$ should correspond to typical initialization schemes (e.g. He et al., 2015) which ensure that input and output activations have roughly the same scale. These schemes use $\sigma^2 \sim 1/(\text{number of inputs})$, and if we consider that there are typically $\sim 100$ input channels, and we typically convolve over a $3 \times 3 = 9$ pixel patch, we obtain $\sigma^2 \sim 0.001$, matching the value we use.

### 3.2.2 Recovering Adam(W) in the high-data limit

In the high-data regime where $\eta/\sqrt{\langle g^2 \rangle} \ll \sigma^2$, the empirically measured AdaBayes learning rate, $\sigma^2_{\text{post}}$, approaches the Adam learning rate (Fig. 3; blue line), so AdaBayes becomes approximately equivalent to Adam(W). To understand this convergence, we can leverage the fixed-point expression in Eq. (20) which accurately models empirically measured learning rates,

$$\lim_{\langle g^2 \rangle \to \infty} \sigma^2_{\text{post}} \approx \frac{\eta}{\sqrt{\langle g^2 \rangle}} \tag{23}$$

so the updates (Eq.18b) become equivalent to Adam updates if we take,

$$\eta = \eta_{\text{Adam}}. \tag{24}$$

Table 1: A table displaying the minimal test error and test loss for a ResNet and DenseNet applied to CIFAR-10 and CIFAR-100 for different optimizers. The table displays the best adaptive algorithm (bold), which is always one of our methods: either AdaBayes or AdaBayes-FP. We also display the instances where SGD (gray) beats all adaptive methods (in which case we also embolden the SGD value).

| | CIFAR-10 | | | | CIFAR-100 | | | |
| | ResNet | | DenseNet | | ResNet | | DenseNet | |
| optimizer | err. (%) | loss | err. (%) | loss | err. (%) | loss | err. (%) | loss |
|---|---|---|---|---|---|---|---|---|
| SGD | 5.170 | **0.174** | 5.580 | 0.177 | **22.710** | **0.833** | **21.290** | **0.774** |
| Adam | 7.110 | 0.239 | 6.690 | 0.230 | 27.590 | 1.049 | 26.640 | 1.074 |
| AdaGrad | 6.840 | 0.307 | 7.490 | 0.338 | 30.350 | 1.347 | 30.110 | 1.319 |
| AMSGrad | 6.720 | 0.239 | 6.170 | 0.234 | 27.430 | 1.033 | 25.850 | 1.103 |
| AdaBound | 5.140 | 0.220 | 4.850 | 0.210 | 23.060 | 1.004 | 22.210 | 1.050 |
| AMSBound | 4.940 | 0.210 | 4.960 | 0.219 | 23.000 | 1.003 | 22.360 | 1.017 |
| AdamW | 5.080 | 0.239 | 5.190 | 0.214 | 24.850 | 1.142 | 23.480 | 1.043 |
| AdaBayes-FP | 5.230 | **0.187** | 4.910 | **0.176** | 23.120 | **0.935** | 22.600 | **0.934** |
| AdaBayes | **4.840** | 0.229 | **4.560** | 0.222 | **22.920** | 0.969 | **22.090** | 1.079 |

As such, we are able to use past experience with good values for the Adam learning rate $\eta_{\text{Adam}}$, to set $\eta$: in our case we use $\eta = 0.001$.

Furthermore, when we consider the form of regularisation implied by our updates, we recover a state-of-the-art variant of Adam, known as AdamW (Loshchilov & Hutter, 2017). In standard Adam, weight-decay regularization is implemented by incorporating an L2 penalty on the weights in the loss function, so the gradient of the loss and regularizer are both normalized by the root-mean-square gradient. In contrast, AdamW "decouples" weight decay from the loss, such that the gradient of the loss is normalized by the root-mean-square gradients, but the weight decay is not. To see that our updates correspond to AdamW, we combine Eq. (18a) and Eq. (18b), and substitute for $\sigma^2_{\text{post}}$ (Eq. 23),

$$\mu_{\text{post}}(t) \approx (\lambda - 1)\, \mu_{\text{post}}(t - 1) + \frac{\eta}{\sqrt{\langle g^2(t) \rangle}} \langle g(t) \rangle. \tag{25}$$

Indeed, the root-mean-square normalization applies only to the gradient of the loss, as in AdamW, and not to the weight decay term, as in standard Adam.

Finally, note that AdaBayes-FP becomes *exactly* AdamW when we set $\sigma^2 \to \infty$,

$$\lim_{\sigma^2 \to \infty} \frac{1}{\sigma^2_{\text{post}}} = \lim_{\sigma^2 \to \infty} \left( \frac{1}{2\sigma^2} + \sqrt{\frac{1}{4\sigma^4} + \frac{\langle g^2 \rangle}{\eta^2}} \right) = \frac{\sqrt{\langle g^2 \rangle}\eta}{,} \tag{26}$$

because we use the standard Adam(W) approach to computing unbiased estimates of $\langle g \rangle$ and $\langle g^2 \rangle$ (see Algo. 2).

## 4   Experiments

For our experiments, we have adapted the code and protocols from a recent paper (Luo et al., 2019) on alternative methods for combining non-adaptive and adaptive behaviour (AdaBound and AMSBound). They considered a 34-layer ResNet (He et al., 2016) and a 121-layer DenseNet on CIFAR-10 (Huang et al., 2017), trained for 200 epochs with learning rates that decreased by a factor of 10 at epoch 150. We used a batch size of 128. We used the exact same networks and protocol, except that we run for more epochs, we plot both classification error and the loss, and we use both CIFAR-10 and CIFAR-100. We used their optimized hyperparameter settings for standard baselines (including SGD and Adam), and their choice of hyperparameters for their methods (AdaBound and AMSBound). For AdamW and AdaBayes, we used $\eta_{\text{SGD}} = 0.1$ and set $\sigma^2$ using Eq. (22), and we used $\eta_{\text{Adam}} = \eta = 0.001$ (matched to the optimal learning rate for standard Adam). We used decoupled weight decay of $5 \times 10^{-4}$ (from Luo et al., 2019), and we used the equivalence of SGD with weight decay and SGD with decoupled weight decay to set the decoupled weight decay coefficient to $\lambda = 5 \times 10^{-5}$ for AdamW, AdaBayes and AdaBayes-FP.

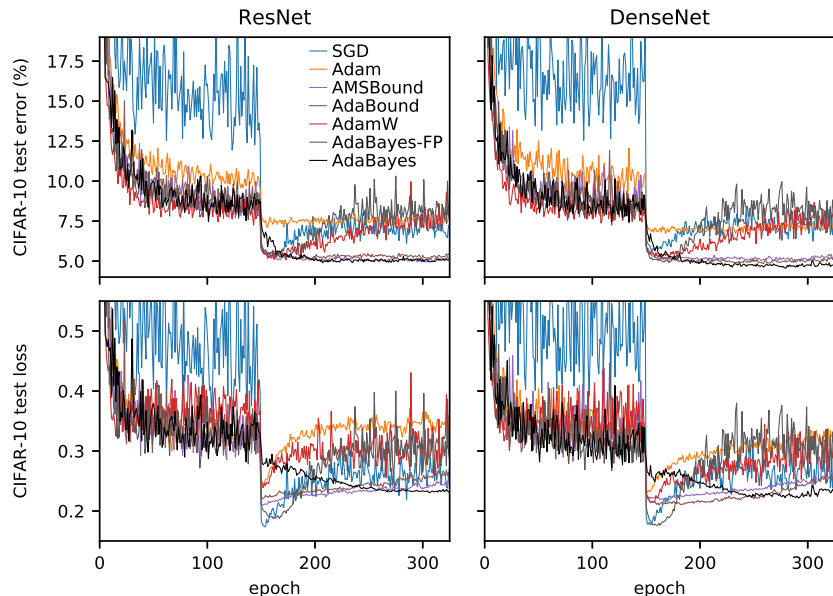

Figure 4: Test loss and classification error for CIFAR-10 for a Resnet-34 and a DenseNet-121, for multiple update algorithms.

The results are given in Table 1 and Fig. 4. The best adaptive method is always one of our methods (AdaBayes or AdaBayes-FP), though SGD is frequently superior to all adaptive methods tested. To begin, we compare our methods (AdaBayes and AdaBayes-FP) to the canonical non-adaptive (SGD) and adaptive (Adam) method (see Fig. A8 for a cleaner figure, including other baselines). Note that AdaBayes and AdaBayes-FP improve their accuracy and loss more rapidly than baseline methods (i.e. SGD and Adam) during the initial part of learning. Our algorithms give better test error and loss than Adam, for all networks and datasets, they give better test error than SGD for CIFAR-10, and perform similarly to SGD in the other cases, with AdaBayes-FP often giving better performance than AdaBayes. Next, we see that AdaBayes-FP improves considerably over AdaBayes (see Fig. A9 for a cleaner figure), except in the case of CIFAR-10 classification error, where the difference is minimal.

Given the difficulties inherent in these types of comparison, we feel that only two conclusions can reasonably be drawn from these experiments. First, AdaBayes and AdaBayes-FP have comparable performance to other state-of-the-art adaptive methods, including AdamW, AdaBound and AMS-Bound. Second, and as expected, SGD frequently performs better than all adaptive methods, and the difference is especially dramatic if we focus on the test-loss for CIFAR-100.

## 5   Conclusions

Our fundamental contribution is to show that, if we seek to use Bayesian inference to perform stochastic optimization, we need a model describing the dynamics of all the other parameters as they are optimized. We found that even by assuming that the other parameters obey oversimplified autoregressive dynamics, we recovered state-of-the-art adaptive optimizers (AdamW). In our experiments, either AdaBayes or AdaBayes-FP outperformed other adaptive methods, including AdamW (Loshchilov & Hutter, 2017), and Ada/AMSBound (Luo et al., 2019), though SGD frequently outperformed all adaptive methods. We hope that understanding optimization as inference, taking into account the dynamics in the other weights as they are optimized, will allow for the development of improved optimizers, for instance by exploiting Kronecker factorisation (Martens & Grosse, 2015; Grosse & Martens, 2016; Zhang et al., 2017).

# 6 Broader Impact

As neural networks are increasingly being used in safety-critical settings such as medical imaging, it is important to ensure that practical neural network optimizers are capable of achieving effective performance in limited training time. We provide a neural network optimization algorithm inspired by Bayesian filtering that is indeed capable of learning rapidly and generalising well. More importantly, by highlighting the connections between Bayesian inference and optimization, we hope to provide a general approach to building new optimization algorithms that will be exploited by future research.

## Footnotes

[1] here we use $x \sim y$ as in Physics to denote "$x$ has the same order of magnitude as $y$", see Acklam and Weisstein "Tilde" MathWorld. http://mathworld.wolfram.com/Tilde.html

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
