[Supplementary Material]

## A  Hessian for minibatches

To confirm that the loss-surface is locally well-approximated by a quadratic function, we plotted the summed minibatch loss for 10 different minibatches (colors) for 20 different randomly chosen directions in parameter space (plots; Fig. A1) for a ResNet trained for 10 epochs. Subtracting the minimum, we find that the loss-functions for different minibatches have a similar shape, and hence have a similar Hessian (Fig. A2). This is expected, because the Hessian for a minibatch is the average of the Hessians for individual datapoints, so as the minibatch size increases, the minibatch Hessian will converge on its expectation. Further, we provide a local quadratic fit, indicating that the minibatch losses are well-approximated by a quadratic (black line; Fig. A2), and hence that the full batch loss is well-approximated by a quadratic. Finally, we plotted the Hessian against the variance of the minibatch gradient, showing a high correlation (Fig. A3)

## B  Gaussianity of minibatch gradients

To confirm the Gaussianity of minibatched gradients, we plotted probability and cumulative density for the minibatch gradients for one element of each weight-matrix and bias. We used minibatches of 100, which is less than the 128 we used in our main experiments, so the minibatched gradients used in the main text should be at least as Gaussian as those displayed here. Further, we standardised the gradients by subtracting the mean and dividing by the standard deviation. We began with a fixed but untrained network, and plotted the empirical histogram of gradients against the probability density for a Gaussian (Fig. A4), and a quantile-quantile plot (the empirical cdf vs the standard Gaussian cdf; Fig. A5). Then, we considered the probability density (Fig. A6) and the quantile-quantile plot (Fig. A7) for a single ResNet trained for 10 epochs. All of these plots indicate that minibatch gradients are well-approximated by a Gaussian distribution.

## C  Mean square normalizer in Ollivier (2017)

In our framework, we can encode the multiplication by the forgetting factor in the computation of $\sigma^2_{\text{prior}}(t+1)$ from $\sigma^2_{\text{post}}(t)$,

$$\frac{1}{\sigma^2_{\text{prior}}(t+1)} = \frac{1-\lambda}{\sigma^2_{\text{post}}(t)}, \tag{27}$$

and the equivalent process noise is,

$$\eta^2 = \frac{\lambda}{1-\lambda}\sigma^2_{\text{post}}(t). \tag{28}$$

To understand the typical learning rates in this model we perform a fixed-point analysis by substituting Eq. (27) into Eq. (17a),

$$\frac{1}{\sigma^2_{\text{post}}(t+1)} = \frac{1-\lambda}{\sigma^2_{\text{post}}(t)} + g^2(t). \tag{29}$$

Solving for the fixed point, $\sigma^2_{\text{post}} = \sigma^2_{\text{post}}(t) = \sigma^2_{\text{post}}(t+1)$, this choice of process noise gives a mean-squre normalizer,

$$\sigma^2_{\text{post}} = \frac{\lambda}{\langle g^2 \rangle}. \tag{30}$$

## D  Kalman filter

The log-posterior (Eq. 15b) is the sum of the log-prior (Eq. 15a) and the log-likelihood (Eq. 11). As such,

$$-\frac{1}{2\sigma^2_{\text{post}}}\left(w^*_i - \mu_{\text{post}}\right)^2 = -\frac{1}{2\sigma^2_{\text{prior}}}\left(w^*_i - \mu_{\text{prior}}\right)^2 - \frac{1}{2H_{ii}}\left(g_i - H_{ii}\left(w^*_i - \mu_{\text{prior}}\right)\right)^2. \tag{31}$$

The quadratic terms allow us to identify $\sigma^2_{\text{post}}$,

$$-\frac{1}{2\sigma^2_{\text{post}}}w^{*2}_i = -\frac{1}{2\sigma^2_{\text{prior}}}w^{*2}_i - \frac{1}{2}H_{ii}w^{*2}_i, \tag{32}$$

Figure A1: Loss function along 20 random directions (plots) for 10 different minibatches (colors).

Figure A2: Loss function along 20 random directions (plots) for 10 different minibatches (colors), with constant offset, and quadratic fitted to all data (black line).

Figure A3: Hessian (based on quadratic fits to full batch loss) plotted against minibatch gradient variance, evaluated at zero perturbation.

so,

$$\frac{1}{\sigma_{\text{post}}^2} = \frac{1}{\sigma_{\text{prior}}^2} + H_{ii}. \tag{33}$$

or,

$$\sigma_{\text{post}}^2 = \frac{1}{\frac{1}{\sigma_{\text{prior}}^2} + H_{ii}}. \tag{34}$$

And the linear terms allow us to identify $\mu_{\text{post}}$,

$$\frac{\mu_{\text{post}}}{\sigma_{\text{post}}^2} w_i^* = \frac{\mu_{\text{prior}}}{\sigma_{\text{prior}}^2} w_i^* + g_i w_i^* + H_{ii} \mu_{\text{prior}} w_i^* \tag{35}$$

so,

$$\mu_{\text{post}} = \sigma_{\text{post}}^2 \left( \left( \frac{1}{\sigma_{\text{prior}}^2} + H_{ii} \right) \mu_{\text{prior}} + g_i \right) \tag{36}$$

identifing $1/\sigma_{\text{post}}^2$,

$$\mu_{\text{post}} = \mu_{\text{prior}} + \sigma_{\text{post}}^2 g_i. \tag{37}$$

Based on Eq. (11) and Appendix A, as $H_{ii}$ is unknown, we use,

$$H_{ii} \approx g_i^2(t). \tag{38}$$

# E    Fixed point variance

For the fixed-point covariance, it is slightly more convenient to work with the inverse variance,

$$\lambda_{\text{post}} = \frac{1}{\sigma_{\text{post}}^2}, \tag{39}$$

though the same results can be obtained through either route. Substituting Eq. (16b) into Eq. (17a) and taking $\eta^2/\sigma^2 \ll 1$, we obtain an update from $\lambda_{\text{post}}(t)$ to $\lambda_{\text{post}}(t+1)$,

$$\lambda_{\text{post}}(t+1) = \frac{1}{\frac{1-\eta^2/\sigma^2}{\lambda_{\text{post}}(t)} + \eta^2} + g^2(t) \tag{40}$$

assuming $\lambda_{\text{post}}$ has reached fixed-point, we have $\lambda_{\text{post}} = \lambda_{\text{post}}(t) = \lambda_{\text{post}}(t+1)$,

$$\lambda_{\text{post}} = \frac{1}{\frac{1-\eta^2/\sigma^2}{\lambda_{\text{post}}} + \eta^2} + \langle g^2 \rangle. \tag{41}$$

Figure A4: Empirical histogram of standardised minibatch gradients compared to the probability density function of the standard normal for an untrained network.

Figure A5: Empirical cumulative density of gradients compared to the cumulative density for a standard normal for an untrained network.

Figure A6: Empirical histogram of standardised minibatch gradients compared to the probability density function of the standard normal for a network trained for 10 epochs.

Figure A7: Empirical cumulative density compared to the cumulative density for a standard normal for a network trained for 10 epochs.

Rearranging,

$$\lambda_{\text{post}} = \frac{\lambda_{\text{post}}}{1 - \eta^2/\sigma^2 + \eta^2 \lambda_{\text{post}}} + \langle g^2 \rangle. \tag{42}$$

Assuming that the magnitude of the update to $\lambda_{\text{post}}$ is small, we can take a first-order Taylor of the first term,

$$\lambda_{\text{post}} \approx \lambda_{\text{post}} \left( 1 + \frac{\eta^2}{\sigma^2} - \eta^2 \lambda_{\text{post}} \right) + \langle g^2 \rangle. \tag{43}$$

cancelling,

$$0 \approx \eta^2 \lambda_{\text{post}}^2 - \frac{\eta^2}{\sigma^2} \lambda_{\text{post}} - \langle g^2 \rangle, \tag{44}$$

and rearranging,

$$0 \approx \sigma^2 \lambda_{\text{post}}^2 - \lambda_{\text{post}} - \frac{\langle g^2 \rangle \sigma^2}{\eta^2}, \tag{45}$$

and finally substituting for $\lambda_{\text{post}}$ gives the expression in the main text.

## F   Additional data figures

Here, we replot Fig. 4 to clarify particular comparisons. In particular, we compare AdaBayes(-FP) with standard baselines (Fig. A8), Adam AdamW and AdaBayes-FP (Fig. A9), AdaBayes(-FP) and Ada/AMSBound (Fig. A10), and Ada/AMSBound and SGD (Fig. A11). Finally, we plot the training error and loss for all methods (Fig. A12; note the loss does not include the regularizer, so it may go up without this being evidence of overfitting).

Figure A8: Test loss and classification error for CIFAR-10 and CIFAR-100 for a Resnet-34 and a DenseNet-121, comparing our methods (AdaBayes and AdaBayes-FP) with standard baselines (SGD, Adam, AdaGrad and AMSGrad (Reddi et al., 2018)).

Figure A9: Test loss and classification error for CIFAR-10 and CIFAR-100 for a Resnet-34 and a DenseNet-121, comparing Adam, AdamW and AdaBayes-FP.

Figure A10: Test loss and classification error for CIFAR-10 and CIFAR-100 for a Resnet-34 and a DenseNet-121, comparing our methods (AdaBayes and AdaBayes-FP) with AdaBound/AMSBound Luo et al. (2019).

Figure A11: Test loss and classification error for CIFAR-10 and CIFAR-100 for a Resnet-34 and a DenseNet-121, comparing AdaBound/AMSBound Luo et al. (2019) and SGD.

Figure A12: Train loss and classification error for CIFAR-10 and CIFAR-100 for a Resnet-34 and a DenseNet-121, for all methods in Fig. 4.