[Reviews · NeurIPS 2020]

Review 1

Summary and Contributions: The paper formulates adaptive and vanilla stochastic gradient descent from the Bayesian inference perspective. The optimization of each particular parameter (i.e. network weight) is formulated as a minimization problem given other weights and correlation between weights. A Gaussian assumption is made, so that the problem can be modeled into a Bayesian inference (Kalman filtering) process, and the optimal learning rate at each step is formulated as the posterior standard deviation of the Gaussian. Through this formulation, SGD and Adam can be recovered under different data assumptions. The proposed method is tested on Cifar-10 and Cifar-100.

Strengths: The Bayesian formulation is interesting to me. Formulating stochastic optimization into estimating the Gaussian posterior seems novel. In the introduction part, the author introduced the relationship between Kalman filtering and natural gradients, which is an interesting topic to explore. The derivation is easy to follow, and recovering existing methods under the Bayesian settings could potentially provide insight for future optimizer design.

Weaknesses: Although the Gaussian formulation does not fully convince me, I am fine with such an assumption. Still. I am expecting the authors to further justify this heuristic setting. My major concern is about insufficient experiments to support the paper's theoretical claims. The paper only reports test error and loss curves on Cifar. To make a solid paper, the authors should make either more detailed analysis on the same dataset or more results on other datasets, or both. Moreover, the results on Cifar does not show significant improvement over existing methods. Neither computational complexity nor time is reported.

Correctness: In my view, the formulation and derivation are mathematically correct. However, I am not fully convinced of the overall Gaussian settings.

Clarity: The writing of the paper is clear and easy to follow. However, the organization can still be improved. For example, Algorithm 1 and 2 are not introduced in the text.

Relation to Prior Work: Yes. Prior works are properly discussed by the paper.

Reproducibility: Yes

Additional Feedback: Post rebuttal: The author feedback addressed my concerns on Gaussian assumptions and computational complexity, and I am convinced of the authors' claims. Therefore I upgrade my rating to marginally below the acceptance threshold. All reviewers agree that a Bayesian interpretation of existing SOTA optimization methods is novel and interesting. However, insufficient experimental analysis is a major weakness of the current manuscript. I agree with R3 on the comment that experiments should be replaced with content that better supports the main argument of the paper instead of simple comparison with existing SOTA methods.


Review 2

Summary and Contributions: The paper introduces a Bayesian Filtering framework to view stochastic optimization methods used in deep learnings such as SGD and Adam, using the derived framework the authors present a new Byesian filtering method named AdaBayes which benefits from good optimization and generalization properties.

Strengths: The work presents AdaBayes and derives it from the Bayesian filtering framework. The proposed AdaBayes optimization method performs well in terms of optimization and generalization. In addition the work does a great job introducing the reader to the previous literature on adaptive methods and Bayesian filtering, and the organization of the paper is quite nice. I enjoyed reading this work.

Weaknesses: Despite the strengths mentioned above the derivation of the Bayesian Filtering framework is not rigorous and is based off of a number unjustified steps. Starting from the setttings of stochastic optimization and Bayesian filtering, multiple reduction steps which include un-realistic assumptions, weaken the connection between the initial Bayesian filtering framework and the derived AdaBayes optimizer. 1.The few sentences in 77-81 are non-rigorous and not well justified. Why should the factorized model of the parameters make sense? 2. The argument that the mini-batch gradients noise follows a normal distribution is a topic of recent research and discussion. 3. In equation 12, the updates on the weights are confusing, why would the parameters of the network be updated according to a constant multiple of their current value? This does not seem to reflect of gradient optimization. Even if sigma is time-varying, I am having a hard time wrapping my head around this. 3. The simplification replacing the Hessian by the squared gradient is non-trivial, and seems to be the casue for the "desired" RMS style optimizer. Finally the introduction of lambda replacing eta/2sigma^2 additionally extends the gap between the resultant optimizer and what we would expect from the Bayesian filtering model. Minor issues: - "philosophical note" paragraph seems a bit digressive. - Line 109 failed to use superscript? - Line 223 Needs proper definition of OU acronym. ___ After reviewing the rebuttal, the authors were able to address some of my concerns, At the same time I find some of the approximations to still not be well justified. I am maintaining my current score for now.

Correctness: Justifications for some of the simplifying assumptions made throughout the work are not sufficient. See weaknesses.

Clarity: The paper is very nice to read, it is well organized and well written.

Relation to Prior Work: Yes, the paper compares to other Bayesian Filtering and adaptive optimization methods in the exposition.

Reproducibility: Yes

Additional Feedback:


Review 3

Summary and Contributions: Post-rebuttal: Dear authors, thank you for your detailed response and offering to fix many points we raised. I would like to sum up my thoughts after having read the other reviews and your rebuttal: On a high level, the following aspects were most significant how I approached towards my final score: 1) The perspective is novel, and has interesting potential. 2) The approximations seem very strong. 3) Experiments are not conclusive. Re 1: I think we all agree that this is a pro for the paper and should be considered its main strength. Re 2: Questioning the approximations is a valid point. However, as you argue, you provided sufficient empirical evidence for the mini-batch Gaussianity, and I think that Gaussianity is often assumed without further justification in other Bayesian inference applications as well, simply to keep the computations tractable. Even if the assumptions are not fully realistic, they seem to be "less concerning than those in past work" (rebuttal, line 19). I appreciate that you are aware of them and honest about the limitations; and that you have further ideas how to improve on them in the future. Re 3: This is the main weakness. I accept that the main contribution of the paper is "understanding" from a different perspective. However, this does not imply that there is no need to show that this knowledge can be "transferred" into practical improvements. I offered to reconsider my score if you provided other experiments that would demonstrate more clearly how your understanding can be used. The original choice (optimizer benchmarking) is sub-optimal to me, and you agree that "these plots aren't the main point" (rebuttal, line 44). Unfortunately, you did not respond to that point, even though I am sure you thought about ways to approach that. I would like to share a "wild" idea that might be a starting point: Maybe you can use the interpolation property of AdaBayes between SGD and Adam to test it on problems where it is known that one of the algorithms fails: - I came across one such example while trying to train a CNN with many (6-8) sigmoids on Cifar10. SGD cannot train such a model (vanishing gradients). Interestingly, Adam seems to be able to. If you used AdaBayes on such a problem, showing that it behaves Adam-like could be a neat use case to illustrate the strengths of the interpolation aspect. - However, I am not aware of a good setting to show the opposite direction; that AdaBayes behaves like SGD in a setting where Adam cannot converge, while SGD does. I would have felt inclined to increase my score if one surprising experiment that shows a clear practical benefit of your insights was provided. As this aspect is still missing in the work, I will stick to my original assessment. ----- This paper establishes interpretations of SGD and Adam-family optimizers from a Bayesian filtering perspective. It introduces a sequential graphical model for neural network optimization, and performs Bayesian inference (filtering) on that model: The optimal value of a single trainable parameter, which changes over time as all parameters are optimized jointly, is inferred from gradient observations. This leads to a new optimizer called AdaBayes. AdaBayes is shown to be connected to both SGD and Adam. In particular, the authors find that the fixed-point of AdaBayes' learning rate adaptation reproduces Adam's root mean square rescaling. Recent approaches have only established an interpretation of rescalings with the mean squared gradient.

Strengths: - Approach: The main goal of this work is to develop a theoretical understanding of adaptive first-order methods from a Bayesian perspective. This is a sane approach: The field of deep learning optimizers is currently facing a rapidly growing number of new proposed methods. To me, it is unclear how significantly those methods differ. A unifying formulation with connections to Bayesian inference may help to improve that situation of the field, and help build significantly better methods in the future. - Established connections: I think that the presented relations between filtering and adaptive first-order optimization are the core strengths of the paper. In particular: - AdaBayes is shown to be connected to SGD and Adam(W). - Adam's root mean square rescaling is derived by taking the large-data limit of the filter's fixed point. - Comparison with other adaptive methods: It is shown empirically that AdaBayes slightly outperforms other adaptive competitors. This is weak evidence for the filtering perspective being promising.

Weaknesses: - First, I would like to mention that I am not very familiar with filtering methods. That said, I would have appreciated a more detailed introduction, since the paper is attempting to build a bridge between optimization and Bayesian filtering. - Inconclusive experiments: As already indicated by the authors, the performed experiments do not yield novel insights. - From my remarks about the current situation in the deep learning optimization community above, I find it unsurprising that a comparison of AdaBayes with other competitors does not yield a clear winner. To me, the most interesting contribution of the paper is the new perspective on existing methods, not the development of a new optimizer. Hence, I think the experiments should be replaced with content that better supports the main argument of the paper. I will reconsider my score if the authors provide more conclusive experimental results. - I have a question about the experimental evaluation: You are recycling the experimental setup of an existing work. Why do you have to extend the training duration? Figure 4 shows that your proposed optimizers are still able to improve after epoch 200, while the competitors already overfit. Do your results change when using 200 epochs as in the original setting?

Correctness: - The experimental setup is borrowed from another paper and hyperparameter choices are recycled for baselines. - I did not entirely follow/understand all mathematical derivations in the main text. Many details are provided in the appendix. From the high-level description provided by the main text, I did not identify a main flaw. - Fisher approximation (weak criticism): The authors correctly mention in their work that using the squared gradient as an approximation to the Hessian is an arguable choice and improving it "is an important avenue for future research". This relation between the squared mean gradient ⟨g²⟩ and the Fisher information/Hessian has lately been critically discussed in [1]. I think it would be a good idea to reference this, or a similar, work to emphasize the potential shortcomings of this approximation, which is highly popular. Understanding its limitations may be essential to improve adaptive optimization methods. Following the same line of the argument, I think that phrases like "as the values of ⟨g²⟩ decreased, corresponding to smaller values in the Fisher Information..." should be phrased more carefully. [1] Kunstner, F., Balles, L., and Hennig, P., Limitations of the empirical Fisher approximation for natural gradient descent (2019).

Clarity: As stated above, I do not have detailed knowledge about filtering methods, which made some derivations hard to follow. Since the paper is building a bridge between two communities, the authors should consider to include more explanations along the derivations, and the used quantities, to improve comprehensibility. For instance, what is σ² in Equ. (12)?

Relation to Prior Work: The work is based on three previous works which obtain mean squared gradient normalizers. I am not aware of the details of these works, but they seem to require unintuitive modifications to derive Adam's root mean square normalizer. In that sense, the paper's contribution is clearly distinguished, as it obtains this normalization more naturally.

Reproducibility: Yes

Additional Feedback: Suggestions: - (Re: prior work) Could you comment why the mean squared normalizer obtained by other works is not desirable for optimization? - Fig. 4: The colors of AMSBound, AdaBound and AdaBayes-FP are hard to distinguish. Please improve the color scheme or use different line styles. - Fig. 4: Results for CIFAR-10 are shown, while the caption mentions both CIFAR-10 and CIFAR-100. - Line 223: Please spell out "OU dynamics" - Section 4: Which batch size is used for training? - Something is wrong in Equ. (26): Instead of "lim 1 / σₚₒₛₜ²" it should be "lim σₚₒₛₜ²" and consequently, the middle term "lim (...)" should be "lim (...)⁻¹" Minor corrections: - Line 1, 25: "cannonical" → "canonical" - Line 109: Broken superscript(s) - Line 129: Missing "be" - Table 1 and line 216: "AdaBayes-SS" → "AdaBayes-FP"


Review 4

Summary and Contributions: *update* I have read the author response, and thank the authors for addressing the issues raised by me. I am still unclear about the 0.1 learning rate for SGD, but hope to see it addressed in the updated manuscript. I am not changing my overall score for the review. --- The paper considers neural network optimization as a Bayesian inference problem, and addresses prior work in this area by considering the temporal dynamics of the parameters as they are optimized. This leads to an intuitive optimizer that interpolates between Adam (a state of the art optimizer) and vanilla gradient descent, and also recovers Adam W — a variant of Adam with decoupled weight decay.

Strengths: * The paper is the first to demonstrate how viewing optimization as Bayesian inference requires modeling temporal dynamics * The proposed algorithm is easy to implement and is computationally efficient

Weaknesses: * The paper makes several “hand-wavy” arguments, which are suitable for supporting the claims in the paper; but it is unclear if they would generalize for analyzing / developing other algorithms. For instance: 1. Replacing `n^2/(2*s^2)` with an arbitrary parameter `lambda` (lines 119-121) 2. Taking SGD learning rate ~ 0.1 (line 164) — unlike the Adam default value, it is unclear what the justification behind this value is.

Correctness: My one concern regarding correctness is if it is trivial to replace gradient with the moving average (line 122). I request the authors to provide justification for this.

Clarity: * The paper suffers from some readability issues, especially with notation and convention. In particular, Equation 12 introduces variable Q without explanation. Equation 6 introduces H but it is not clarified to be the Hessian till later. * Figure 2 would be more useful if it showed how mu and sigma interact with the other parameters. * The plots in Figure 4 are very cluttered — I suggest plotting losses “less frequently” rather than every epoch, to get smoother curves.

Relation to Prior Work: Yes, the paper is clear about how it is the first to consider the temporal dynamics when considering optimization as an inference problem.

Reproducibility: Yes

Additional Feedback:

[Author Response · NeurIPS 2020]

Our work "establishes interpretations of SGD and Adam-family optimizers from a Bayesian filtering perspective" (R3). It is "the first to demonstrate how viewing optimization as Bayesian inference requires modeling temporal dynamics" (R4) and results in an algorithm that is "easy to implement and is computationally efficient" (R4) and "benefits from good optimization and generalization properties" (R2). Finally, the reviewers recognised the potential impact of our method "A unifying formulation with connections to Bayesian inference may help to improve that situation of the field, and help build significantly better methods in the future" (R3)

**Shared points. Gaussianity (R1, R2).** We included considerable empirical analysis of the Gaussianity in appendices A+B, and Figs A1-7. All show excellent empirical agreement with Gaussianity in this setting. Note, this is partly because we care about minibatch gradients, which are averages of 128 independent training-example-gradients, so central-limit arguments push the distribution towards Gaussianity (but without taking the limit, all we can do is establish empirical agreement). **Empirical results (R1, R3).** While our method improves over all baseline adaptive methods, and often also SGD, we agree that these improvements are not spectacular. Our major contribution is to give a Bayesian approach that "interpolates between Adam (a state of the art optimizer) and vanilla gradient descent, and also recovers Adam W" (R4), and therefore explains the excellent performance of these SOTA methods. We would hope that our rigorous approach would give some performance improvements, which it does, but the underlying similarities to these SOTA methods (which become exact in various asymptotic limits) imply that we cannot expect huge performance differences. **Approximations (R2, R4).** Inevitably, there are approximations and heuristics (including $\eta/(2\sigma^2)$) required to obtain an efficient and effective method in these extremely challenging high-dimensional settings. However, these are much less concerning than those in past work (Khan et al 2018) that required "unintuitive modifications to derive Adam's root mean square normalizer"(R3). We believe that these issues can be resolved by using a more complex dynamical prior over weights. However, this approach introduces considerable additional complexity, which is simply too much for the first paper "to demonstrate how viewing optimization as Bayesian inference requires modeling temporal dynamics"(R4). **Conclusion.** I would urge the reviewers to consider the value of the approach broadly, and in particular that "A unifying formulation with connections to Bayesian inference may help to improve that situation of the field, and help build significantly better methods in the future." (R3). For instance, one particularly exciting extension is to infer a posterior over a full weight matrix, rather than each element separately. This results in a K-fac variant of Adam, where we precondition updates by the *square root* of the inverse Fisher Information, rather than the inverse Fisher Information as in standard natural gradient approaches. This approach can be expected to offer big benefits in terms of stability and generalisation error, just as Adam, with a RMS gradient normaliser, offers big benefits over using a squared-gradient normaliser. However it is difficult to ask my students to work on these exciting and challenging extensions when this foundational work remains unpublished.

**R1. 3** See "Gaussianity" above. See A8-12 for much more in-depth analysis of the results, including training losses and accuracies, also see "Empirical results" above. We have updated the paper to include a discussion of computational and time complexity, both are $\mathcal{O}(N)$, where $N$ is the number of parameters. Practically, performance is very similar to standard methods such as Adam. **5** We have updated the manuscript to introduce Algos 1 and 2.

**R2. 3.1** In the ideal case you shouldn't use a factorised model, and 77-81 aren't trying to motivate a factorised model. But the high-dimensionality of typical CNNs *forces* approximate, factorised models. 77-81 are only arguing that if we must use a factorised model, we should use one with dynamics. Also, see "Conclusions" above for non-factorised future work. **3.2** See "Gaussianity" above. **3.3** Eq 12 should not yet reflect gradient-based optimization, as it only describes the prior distribution under which we perform inference. The multiplicative decay is necessary because if we just had noise, it would imply that a-priori, the weights slowly grow to infinity. **3.4** The Hessian substitution is standard in the literature (e.g. Khan et al. 2018), but we agree that its improvement is an important avenue for future research. **3.5** See "Approximations" above. **Minor** 1. Agreed, but a few people get very confused on this point. 2. Fixed. 3. Fixed.

**R3. 3.1** We have written a new section introducing filtering methods. **3.2** We agree, these plots aren't the main point, but it remains is valuable to show that our method indeed achieves somewhat improved performance (see "Empirical results" above). **3.3** Many of our existing plots support the main argument of the paper: we have detailed plots showing the Gaussianity assumptions of the method hold (Fig A1-7) and showing that our steady-state limits hold in practice (Fig 3). **3.4** We did not need to run for longer, but it is useful because it gives strictly more information about the performance of the method. The results do not change markedly if we run for 200 epochs. **4** Thanks! We have cited [1] and been more careful about the "FI" terminology. **5+8** Thanks! A number of great points, all fixed.

**R4**. **3.1** See "Approximations" above. **3.2** 0.1 is the default initial learning rate for SGD in these models/datasets, and is the best for SGD in the hyperparameter search. **4.** Excellent point: the momentum is not trivial. We intend to address this in future work, either by introducing a more complex generative model (see "Approximations" above), but doing this rigorously is too complex for this first paper. **5.1** We have cleared up these readability issues especially regarding **Q** and **H**. **5.2** We have added some additional plots showing how $\mu$ (especially changes in $\mu$) interact with $\sigma$. **5.3** We have decluttered Fig 4 by plotting performance every 5 epochs.

[Meta-Review · NeurIPS 2020]

After a discussion with the reviewers, I converged towards recommending to accept this submission. The reviewers raised the following aspects: 1) The perspective is novel, and has interesting potential. 2) The approximations seem very strong. 3) Experiments are not conclusive. Re 1: all reviewers agree that this is a pro for the paper and should be considered its main strength. The authors agree (rebuttal, lines 23-25). Re 2: R3 believes that questioning the approximations is a valid point. However, as the authors argue, they have provided sufficient empirical evidence for mini-batch Gaussianity in appendix B, and Gaussianity is sometimes assumed without further justification in other Bayesian inference applications as well, simply to keep the computations tractable. Even if the assumptions are not fully realistic, they seem to be "less concerning than those in past work" (rebuttal, line 19). R3 appreciated that the authors were aware of them and honest about the limitations of their approach, and that they have further ideas how to improve on them in the future. Re 3: This is the main weakness. R3 accepts that the main contribution of the paper is "understanding" from a different perspective. However, this does not imply that there is no need to show that this knowledge can be "transferred" into practical improvements. R3 shared their ideas with the authors in the updated review. There is no consensus whether point 3 not being addressed outweighs the submission's "interestingness" to the community, in a way that others could build on these insights in the future. Further, all "reject" reviewers indicated low confidence in their assessment. Given the above, I decided to recommend to accept this submission.